# Natural Compounds: Potential Therapeutics for the Inhibition of Cartilage Matrix Degradation in Osteoarthritis

**DOI:** 10.3390/life13010102

**Published:** 2022-12-30

**Authors:** Omer S. Ashruf, Mohammad Yunus Ansari

**Affiliations:** 1Department of Anatomy and Neurobiology, Northeast Ohio Medical University, 4209, State Route 44, Rootstown, OH 44272, USA; 2College of Medicine, Northeast Ohio Medical University, 4209, State Route 44, Rootstown, OH 44272, USA; 3Musculoskeletal Research Focus Area, Northeast Ohio Medical University, 4209, State Route 44, Rootstown, OH 44272, USA

**Keywords:** natural compounds, polyphenols, cartilage matrix, cartilage homeostasis, MMP-13, ADAMTS5, curcumin, olive oil, resveratrol, osteoarthritis

## Abstract

Osteoarthritis (OA) is the most common degenerative joint disease characterized by enzymatic degradation of the cartilage extracellular matrix (ECM) causing joint pain and disability. There is no disease-modifying drug available for the treatment of OA. An ideal drug is expected to stop cartilage ECM degradation and restore the degenerated ECM. The ECM primarily contains type II collagen and aggrecan but also has minor quantities of other collagen fibers and proteoglycans. In OA joints, the components of the cartilage ECM are degraded by matrix-degrading proteases and hydrolases which are produced by chondrocytes and synoviocytes. Matrix metalloproteinase-13 (MMP-13) and a disintegrin and metalloproteinase with thrombospondin motifs 4 and 5 (ADAMTS5) are the major collagenase and aggrecanase, respectively, which are highly expressed in OA cartilage and promote cartilage ECM degradation. Current studies using various in vitro and in vivo approaches show that natural compounds inhibit the expression and activity of MMP-13, ADAMTS4, and ADAMTS5 and increase the expression of ECM components. In this review, we have summarized recent advancements in OA research with a focus on natural compounds as potential therapeutics for the treatment of OA with emphasis on the prevention of cartilage ECM degradation and improvement of joint health.

## 1. Introduction

Osteoarthritis (OA) is the most common degenerative joint disease affecting millions of people worldwide with increasing prevalence with the growth of the aging population. The disease involves pathological changes in the whole joint including cartilage, bone, meniscus, infrapatellar fat pad and synovium which promote the degradation of articular cartilage extracellular matrix (ECM), mainly type II collagen and aggrecan, and reduced cartilage ECM synthesis eventually leading to joint dysfunction and disability [1]. The most common risk factors for the development of the disease include aging, trauma, sex, and obesity [2]. The clinical symptoms of OA include joint pain, swelling, and morning stiffness. There is no disease-modifying drug available for the treatment of OA. An ideal drug candidate is expected to prevent cartilage ECM degradation and promote the synthesis of degraded ECM to restore joint function.

The articular cartilage is a highly specialized connective tissue comprised chondrocytes which occupy less than 5% of the space while more than 95% of the cartilage is occupied by cartilage ECM. Chondrocytes are the primary cell type present in the cartilage and are responsible for the maintenance of the ECM. The cartilage ECM is composed of a complex network of biological macromolecules that exist in an aqueous environment that allows nutrients, growth factors, and cytokines to move freely between chondrocytes and synovial fluid [3,4]. Together the chondrocytes and pericellular matrix constitute the chondrons, the functional unit of cartilage [5]. The biological macromolecules of ECM predominantly include type II collagen fibers, which make the structural backbone, and proteoglycan, mainly aggrecan, which interacts with the network of collagen fibers and hyaluronic acid to form a macromolecular aggregate [3]. In addition to type II collagen, the cartilage ECM also has smaller quantities of type I, IV, V, VI, IX, and type XI collagens and other proteoglycans such as decorin, biglycan, and fibromodulin in minor quantities [6,7]. The cartilage ECM also contains some non-collagen and non-proteoglycan proteins such as fibronectin and laminin which are believed to provide structural support to the cartilage.

Matrix metalloproteases (MMPs) are a family of metal-dependent proteases and are highly expressed in OA cartilage [8,9]. They have a variety of roles in cartilage metabolism but are specifically implicated in ECM degradation as collagenases and proteolytic enzymes. Specifically, MMP-1, MMP-3, and MMP-13 are hallmarks of OA progression, with MMP-13 showing a strong affinity to the degradation of type II collagen [10,11,12]. They are produced by the joint tissue including chondrocytes, synoviocytes, and meniscus in response to inflammatory cytokines and mechanical stress [13,14]. Targeting the expression of MMPs in the cartilage of OA models has shown to be a therapeutic strategy, underscoring the importance of studying compounds that inhibit their release [10,12]. Interestingly, mice expressing active human MMP-13 in cartilage show OA-like symptoms similar to those observed in humans [15]. It was reported that cartilage-specific, conditional deletion of Mmp-13 from a mouse model reduced the progression of surgically induced OA and improved cartilage matrix synthesis [12]. Pharmacological inhibition of Mmp-13 reduced the severity of the disease in mouse and rat models [12,16]. The results from these animal studies demonstrate that MMP-13 is a critical player in OA pathogenesis, suggesting that MMP-13 inhibition may reduce disease severity.

The ADAMTS (a disintegrin and metalloproteinase with thrombospondin motifs) family is a class of metal-dependent aggrecanases, responsible for the cleavage of proteoglycan components in cartilage remodeling. ADAMTS4 and ADAMTS5 are of specific importance in OA as they are the major aggrecanases and degrade the proteoglycans in cartilage ECM [17,18]. The role of different ADAMTS in OA is not yet fully understood. In animal model studies, in ADAMTS4 knockout mice, limited effect was seen on the preservation of aggrecan, while there was marked protective effect on aggrecan in the ADAMTS5 knockout counterparts [19,20,21]. Whereas in humans, both ADAMTS4 and ADAMTS5 were reported to play a pathological role in OA [18,22]. Similar to MMPs, pro-inflammatory cytokines and mechanical stress, and mitochondrial dysfunction trigger a cascade of downstream events causing an increase in the expression and activity of ADAMTS5 [8]. The importance of ADAMTS5 in OA pathogenesis can be understood by the fact that the deletion of Adamts5 reduced the cartilage matrix degradation and severity of OA in a mouse model of surgically induced OA [21]. Targeting ADAMTS5 using monoclonal antibodies [23,24,25,26] and small molecules [27,28,29,30] has been shown to prevent cartilage degradation in animal models; however, there are several side effects of these antibody treatments [17]. Based on these results, ADAMTS5 represents a potential target for OA treatment.

Other proteases and hydrolases involved in OA pathogenesis include cathepsin B, cathepsin S, and cathepsin K. The cathepsin family of proteases is a group of cysteine endopeptidases involved in cartilage ECM degradation and are increased in OA cartilage. Generally, they function to complex with glycosaminoglycan components and cleave proteoglycan and other ECM basement proteins. Cathepsin B and S exhibit aggrecanase activity, with cathepsin B cleaving aggrecan near the MMP cleavage site [31,32]. Cathepsin K, which is implicated in OA, and other arthritic diseases and osteoporosis, cleaves type II collagen at multiple sites [33,34]. Its clinical utility has been recognized, as cathepsin K inhibitors are indicated as a possible treatment for OA [35,36].

Natural compounds have been shown to inhibit cartilage ECM degradation by targeting the expression and activity of ECM-degrading proteases and suppressing pain, disease progression and severity in various in vitro, animal models and clinical trial studies. In this review we have discussed the role of matrix degrading proteases, primarily focusing on MMP-13 and ADAMTS5 in cartilage ECM degradation and OA pathogenesis, the major signaling pathways involved and the role of plant derived natural compounds in suppressing cartilage ECM degradation and promoting ECM synthesis. 

## 2. Matrix Degradation Is Central to Osteoarthritis

Type II collagen and aggrecan are the critical components of articular cartilage ECM [6,37]. While collagen provides the fibrous framework and tensile strength, aggrecan provides resistance against compressive loading to the cartilage. Aggrecan is a proteoglycan molecule with abundant side chains of chondroitin sulfate and keratan sulfate and interacts with hyaluronic acid. The cartilage matrix is a dynamic structure and in normal cartilage, the health of the ECM is maintained by a balance between synthesis and degradation. The cleavage of aggrecan in the human OA cartilage occurs at a specific site called the aggrecanase site by ADAMTS family members prompting the release of aggrecan fragments in the synovial fluid. Accumulation of aggrecan fragments in the synovial fluid is a pathological feature associated with cartilage degradation and it serves as an indicator of disease severity [37]. The catabolic changes in the OA joints culminate in increased expression of collagenases and aggrecanases inducing cartilage ECM degeneration, which is a hallmark of OA pathogenesis (Figure 1). In OA joints, there is an imbalance between the synthesis and degradation of cartilage ECM components [38]. The proinflammatory cytokines, TNFα, IL-1β, and IL-6 induce the expression of several MMPs, ADAMTSs, cathepsins, and inflammatory mediators such as COX-2, iNOS, and PGE2 in primary human chondrocytes and synoviocytes [39,40,41,42,43]. These inflammatory mediators also generate radical oxygen species (ROS) which directly contribute to ECM degradation; high levels of ROS cleave collagen and aggrecan, leaving the degradation products in the synovial fluid to further beget inflammation and protease production [39,44]. The knockout mice lacking MMP-13 or ADAMTS5 were found to be protected from the development of experimental OA [12,21]. In addition to the increased expression of MMPs and ADAMTSs inducing cartilage ECM degradation, the synthesis of aggrecan and type II collagen is also downregulated in OA cartilage and chondrocytes [45].

## 3. Signaling Pathways Regulating Matrix Degradation in Osteoarthritis

OA is characterized by a complex meshwork of inflammatory cytokines and mediators which work in concert to induce cartilage ECM degradation (Figure 2). In response to mechanical stress, chondrocytes and other tissues of the joint synthesize key proinflammatory cytokines such as IL-1β, TNFα, and IL-6 early in OA [41,46,47]. These cytokines have been the basis of many studies, often used to replicate the effects of OA on different models [41,48,49]. One of the most important pathways activated in OA and discussed in this review is NFκB signaling [50]. NFκB undergoes nuclear translocation, binding to DNA regulatory elements, and shifting the chondrocyte phenotype to a degradative one [51]. This thereby increases the release and expression of matrix degradative enzymes MMP-1, MMP-9, MMP-13, ADAMTS4, and ADAMTS5 [43]. NFκB further upregulates the same key cytokines IL-1β, TNFα, and IL-6, perpetuating an ECM degradation positive feedback loop [52]. Inhibitors of NFκB have been pursued as potential therapeutic agents of OA, highlighting their importance both mechanistically and clinically [53]. 

Wnt/β-Catenin is another pathway upregulated in OA. Wnt is a glycoprotein that resides in the extracellular compartment. When bound to its specific receptor, Wnt activates Frizzled protein and low-density lipoprotein receptor-associated protein [54]. The release of these proteins inhibits the destruction of β-Catenin from other complexes, allowing it to translocate and accumulate in the nucleus [55]. It has been seen that baseline Wnt activity is essential for chondrocyte metabolic homeostasis, increased activity, as seen in OA, can result in cartilage matrix degradation by associated ECM-degrading proteases [56]. In β-Catenin overexpressing mice, the expression of Mmp-13, Adamts4, and Adamts5 was highly upregulated, whereas cartilage thickness was decreased, and ECM degradation was increased [57]. Wnt/β-Catenin inhibition was shown to reduce cartilage catabolism and lessen OA disease severity in a mouse model [58].

Among these pivotal signaling pathways exists MAPK, a family of serine-threonine protein kinases, which may be activated by several factors, including but not limited to physical stress, infection, inflammation, and extracellular signals. MAPKs have a long list of downstream targets and proteases. All of these targets may induce transcription of regulatory genes, but p38 is of special importance due to its mention in the literature and role as a pathway that also induces translation and stabilized mRNA to do so [59,60]. Phosphorylation cascades are propagated sequentially by kinases MAPKKK (TAK, MLK3, and ASK), MAPKK (MKK3, MKK6), and MAPK in order to activate p38 [61]. Induction of MAPK-p38 yielded MMP-1 and MMP-13 release, degrading collagen, while also increasing IL-1β and TNFα levels [59]. Inhibition of the MAPK-p38 system led to decreased expression of MMP-3 in multiple models [62,63].

Although the interplay of biochemical pathways in OA is incredibly nuanced and varies in quantity, with regards to ECM degradation and the targets of the bioactive compounds discussed below, NFκB, Wnt/β-Catenin, and MAPK signaling represent the most relevant.

## 4. Natural Compounds, Suppressors of Cartilage Matrix Degradation

Flavonoids are bioactive natural compounds that have anti-inflammatory properties, antioxidant activity, and inhibit cartilage extracellular matrix degradation (Figure 2). Matrix metalloproteases and ADAMTSs are highly upregulated in the OA joints and responsible for cartilage ECM degradation [42,44]. Several studies have shown beneficial effects of natural compounds in OA (Table 1). We have shown that a Butein-rich extract from the flowers of *Butea monosperma* and purified Butein was found to suppress the expression of MMP-3, -9, and -13 [64,65]. In addition to flavonoids, several other classes of natural compounds have been extensively studied in the context of OA including isoflavones [66,67], terpenes [68,69], alkaloids [70].

**4.1** Green tea polyphenol, Epigallocatechin gallate (**EGCG**), suppressed MMP-13 expression in advanced glycation end products (AGEs) stimulated human OA chondrocytes [71]. The suppressive effect of EGCG was due to the inhibition of p38- and JNK-MAPK pathways. EGCG also inhibited NFκB activation in primary human OA chondrocytes [71]. An aqueous extract of Java tea (*Orthosiphon stamineus*) suppressed cartilage ECM degradation in cartilage explants and a monosodium iodoacetate (MIA) induced rat model of OA [120]. EGCG was reported to increase the thermal stability of the cartilage and decreased the release of glycosaminoglycans in a cartilage explant study [75]. Intraarticular injection of EGCG in an anterior cruciate ligament transection rat model of OA showed a significant reduction in cartilage degradation by increasing autophagy and reducing MMP-13 expression [72]. Intraperitoneal injection of EGCG in a destabilization of medial meniscus (DMM) surgery-induced OA mouse model was shown to reduce the expression levels of MMP-13 and ADAMTS5 and protected the cartilage extracellular matrix from degradation after four and eight weeks of surgery [74]. These mice also showed reduced OA related pain. Intraarticular injection of EGCG was reported to reduce the severity of OA in a guinea pig model of spontaneous OA by increasing the expression of type II collagen and aggrecan and decreasing the expression of MMP-13 [73].

**4.2 Olive oil** is rich in several natural compounds including hydroxytyrosol, tyrosol, oleocanthal, and oleuropein, and has been shown to suppress cartilage ECM degradation and improve joint health in animal models [121]. Administration of water extract of olive leaves through drinking water increased cartilage healing in a rabbit model of cartilage injury [122]. A combination of exercise and extra-virgin olive oil supplemented diet was shown to improve cartilage health in a rat model of OA by upregulating the expression of lubricin [123]. Hydroxytyrosol functions as a potent antioxidant, preventing the formation of ROS, and its oral administration was shown to suppress IL-1β-induced expression of MMP-13 and protected aggrecan degradation in a mouse and rabbit model of surgically induced OA [76]. In primary human articular chondrocytes, procyanidins, hydroxytyrosol supplementation reduced expression of many inflammatory mediators, such as TNF-α, MMP-3, and PGE2, and exhibited anti-IL-1β expression at different dosages [124].

Although there is limited study on the effect of oleocanthal on matrix-degrading proteases, a study of oleocanthal treatment on lipopolysaccharide-induced OA on human chondrocyte models sheds light on the topic [68]. Specifically, oleocanthal drastically reduced both MMP-13 as well as ADAMTS5 expression in primary human OA chondrocytes. This effect was mediated by oleocanthal-induced inhibition of the MAPK/NFκB pathway and its downstream activation of many other inflammatory cytokines such as IL-6, IL-8, COX-2, and NO [77].

Oleuropein is the most prevalent bioactive compound of olive oil, and derives much of its potency from its metabolite, hydroxytyrosol [125]. Oleuropein inhibited IL-1β-induced activation of NFκB and MAPK pathways and reduced MMP-1, MMP-13, and ADAMTS5 expression, and also inhibited cartilage matrix degradation in primary human OA chondrocytes [78]. Oleuropein suppressed connexin43 activity in OA chondrocytes and restored matrix protein synthesis, by increasing the deposition of COL2A1 and proteoglycan, while also reducing matrix protein degradation [126]. In a guinea pig model of spontaneous OA, an oleuropein or rutin-enriched diet significantly suppressed cartilage degradation and OA progression [80]. Oral administration of oleuropein (50 mg) in a randomized clinical trial including 124 subjects showed low pain in a subset of patients with high pain at the beginning of the treatment [79].

**4.3 Curcumin,** found in turmeric is one of the most studied natural compounds. Curcumin has strong anti-inflammatory and antioxidant activity and has been reported to have chondroprotective activity using various in vitro and in vivo models [127,128,129,130]. Curcumin suppressed the expression of MMP-13 and increased the expression of type II collagen by inhibiting the NFκB pathway in rat primary chondrocytes [85]. Curcumin treatment reduced MMP-3 levels and inhibited aggrecan degradation in the equine cartilage explant model [81,82]. In another study, curcumin was found to suppress IL-1β induced MMP-3 expression in human cartilage explant [83]. Oral gavage administration of curcumin was found to increase the expression of type II collagen and decrease the expression of MMP-8 and MMP-13 in a rat model of zymosan-induced OA [84]. Oral administration of curcumin was found to reduce the expression of Mmp-13 and Adamts5 and increased the expression of chondroprotective transcription factor CITED2 and slowed the progression of disease in a DMM-induced OA mouse model [86]. The application of topical curcumin nanoparticles was found to reduce OA-related pain in the mice [86]. In a 3-month human clinical trial study, supplementation of bio-optimized curcumin capsules to OA patients reduced knee pain, serum levels of COL2A1 degradation products and improved global assessment of OA activity and severity score [131].

**4.4 Honey** has long been part of human history and has cultural significance as a therapeutic agent dating back to before modern civilization. Honey, which is a viscous sugar solution, is also high in natural compounds with therapeutic actions. Honey has a wide range of clinical applications such as wound healing, dermatological issues (ulcerations, eczema, psoriasis), gut microbiome improvement, and hormone signaling in oncogenesis [132,133].

Of special importance is chrysin, a natural flavonoid compound found in honey that has strong anti-inflammatory properties. Human OA chondrocytes treated with chrysin showed a significant decrease in the degradation of aggrecan and type II collagen and reduction in IL-1β induced MMP-1, -3, and -13 and ADAMTS5 expression via NFκB inhibition [87]. In another study, chrysin was shown to reduce high-mobility group box chromosomal protein (HMGB-1) expression in human OA chondrocytes leading to decreased MMP-13 levels [134]. In addition to chrysin, other flavonoids found in honey, such as Fisetin, butein, and luteolin may also serve as a countermeasure to ECM degradation as reported in various in vitro and in vivo studies [64,88,89,132]. 

**4.5 Resveratrol** is a polyphenolic compound found in grapes and has been shown to have chondroprotective activity in various pre-clinical and clinical studies [135,136]. Intraarticular injection of resveratrol was reported to have protective effects in the anterior cruciate ligament transection (ACLT) induced rabbit model of OA [137]. In addition, resveratrol also suppressed AGEs-induced expression of MMP-13 by inhibiting the JNK/ERK-AP-1 pathway and blocked the degradation of type II collagen in a porcine cartilage explant model [138]. Resveratrol was shown to increase the expression of type II collagen and suppress the expression of MMP-1, -3, and -13 and ADAMTS4 and ADAMTS5 in rabbit chondrocytes through the inhibition of the NFκB pathway [91]. Intraarticular injection of resveratrol increased the expression of aggrecan and type II collagen and suppressed the expression of MMP-13 and ADAMTS5 in a DMM-induced OA mouse model by inducing autophagy through AMPK/mTOR signaling pathway [93]. In another study, intraarticular injection of resveratrol activated SIRT1 and suppressed IL-1β-induced expression of HIF-2α in human chondrocytes and slowed the progression of experimental OA in a mouse model of OA [92]. Oral administration of resveratrol prevented the development of high-fat diet-induced OA in the C57BL6 mouse model [90]. In another study, resveratrol was shown to slow the progression of OA in a diabetic mouse model [139]. Clinical trials of resveratrol supplementation have been less conclusive; in a randomized clinical trial of 110 OA patients, 55 were given meloxicam with adjuvant oral 500mg resveratrol once daily for 3 months and 55 were given meloxicam with placebo. Using Western Ontario and McMaster Universities Osteoarthritis Index (WOMAC) scoring, the treatment group saw significant clinical improvement in pain, stiffness, and physical function [140]. However, another clinical trial with similar parameters (50 administered meloxicam with 150mg resveratrol, 32 administered meloxicam with placebo) noted no significant change in clinical relief via WOMAC scoring. Further, the same study found differences in IL-1β, TNF-α, and IL-6 serum level were weakly correlated between the two groups [141]. 

**4.6 Zingerone,** an active ingredient in cooked ginger, has strong anti-inflammatory, antioxidant and chondroprotective activity [142]. Treatment of cartilage explants and chondrocytes with zingerone suppressed IL-1β-induced MMP-13 expression and inhibited cartilage ECM degradation via the suppression of the p38- and JNK-MAPK signaling pathway [94]. In a randomized clinical trial study with 120 OA patients in the age range of 50–75 years, topical application of zingerone in nanostructure lipid carrier showed a significant reduction in OA-related pain in 67% of patients [143]. In other clinical studies, oral administration of ginger powder supplementation, ginger oil massage, or ginger extract in a gel preparation was found to be effective in ameliorating joint pain in OA patients [144,145,146,147]; however, further studies with higher sample size are required to determine its clinical significance.

**4.7 Kaempferol**, a natural compound found in many fruits was shown to suppress IL-1β-induced expression of MMP-1, -3 and -13, and ADAMTS5 in rat chondrocytes via the inhibition of the p38/ERK-MAPK pathway [95]. Kaempferol was also reported to inhibit the degradation of type II collagen [95]. In an in vitro study, kaempferol was found to suppress the levels of STAT3 by upregulating the expression of miR-130a and suppressing cartilage ECM degradation [148]. In another study, kaempferol was found to suppress the expression of inflammatory mediators in rat chondrocytes [149]. In a randomized, double-blind, active-controlled, and parallel-group clinical trial study, an extract from *Elaeagnus angustifolia* enriched in Kaempferol was reported to alleviate pain and improved WOMAC, visual analog scale (VAS), Leguesne’s Pain-Function Index (LPFI), and Patient’s Global Assessment (PGA) scores in OA patients [96].

**4.8 Emodin** is a naturally occurring anthraquinone found in various plants and fungi and has been shown to have antioxidant and anti-inflammatory activities [150]. Emodin treatment reduced the expression of MMP-3 and -13 and AMDATS4 and ADAMTS5 in IL-1β stimulated rat chondrocytes through the inhibition of NFκB and Wnt/β-catenin signaling pathway [97,98]. Emodin treatment also prevented the loss of aggrecan and type II collagen expression in rat chondrocytes [97]. Intraarticular injection of Emodin ameliorated the progression of ACLT-induced experimental OA in a rat model [97].

**4.9 Carnosol** has been shown to have anti-inflammatory and chondroprotective activities [151]. Carnosol treatment downregulated the expression of catabolic mediators of cartilage degradation MMP-3, ADAMTS4, and ADAMTS5 in primary human OA chondrocytes [99]. Carnosol was also shown to increase the expression of cartilage ECM component, type II collagen and aggrecan [99]. 

**4.10 Ferulic acid** treatment of primary human OA chondrocytes was shown to suppress the expression of MMP-1, -3, and -13, and increased the expression of type II collagen and aggrecan by activating SIRT1/AMPK/PGC-1α signaling pathway [101]. Ferulic acid also suppressed the expression of MMP-1 and MMP-13 and restored the expression of SOX9 in porcine chondrocytes [100]. In another study on human OA chondrocytes, ferulic acid suppressed the expression of MMP-1, upregulated the expression of tissue-specific inhibitor of metalloproteinase-1 (TIMP-1) and reduced the severity of experimental OA in a rat model [102].

**4.11 Chlorogenic acid** was found to suppress the expression of MMPs in IL-1β stimulated rabbit chondrocytes through the inhibition of NFκB [103]. Chlorogenic acid treatment was shown to prevent cartilage degradation and slowed the progression of ACLT-induced OA in rabbits [103]. Chlorogenic acid-enriched butanol extract of WIN-34B suppressed the IL-1β induced expression of MMP-1, -3, -13, ADAMTS4, and ADAMTS5 in chondrocytes and reduced the release of glycosaminoglycan and type II collagen [104]. WIN-34B also upregulated the expression of aggrecan and type II collagen in human chondrocytes and cartilage explants [104]. Chlorogenic acid-enriched extract of Anthriscus sylvestris leaves suppressed the expression of MMP-3, -13, and ADAMTS4 and slowed the progression of disease in a rat model of OA [152]. 

**4.12 Quercetin** is a natural compound found in onion that has been shown to have chondroprotective activity. In a study, the combination of Quercetin and vitamin C when given intraarticularly did not improve the OA condition in a rat model [153]. However, intraarticular injection of Quercetin alone mixed in a thermosensitive hydrogel suppressed the cartilage degradation and slowed the progression of OA in a rat model [107]. In a randomized, double blind, clinical study, oral administration of Quercetin combined with glucosamine and chondroitin sulfate for 16 weeks improved cartilage ECM synthesis and decreased the symptoms of knee pain in patients with symptomatic OA, specifically in walking and ascending/descending the stairs, when compared to the control group [106]. In another study, a formulation of Quercetin and palmitoylethanolamide was found to suppress the expression of MMP-1, -3, and -9 in a rat model of OA [105]. 

**4.13 Morin** was reported to suppress the expression of MMP-3 and -13 and upregulated the expression of TIMP-1 in IL-1β stimulated human chondrocytes [108]. Moreover, oral administration of Morin slowed the progression of ACLT-induced OA in a rat model [108].

**4.14** In an in vitro study using a human cartilage explant model, the **B serrata** extract was reported to suppress the cartilage matrix degradation through the inhibition of MMP-9 and -13 expression [109]. A formulation of acetyl-11-keto-β-boswellic acid enriched B serrata extract with non-volatile oils of B serrata gum resin (Aflapin) increased the production of glycosaminoglycans in primary human chondrocytes [154]. The analysis of 11 randomized clinical trials found that B serrata extract significantly reduced pain and improved joint function but in each study, the number of participants was less than 100 and the quality of the study was overall low [155]. 

**4.15** French maritime pine bark extract (**Pycnogenol**) has strong anti-inflammatory, antioxidant and chondroprotective effects in vitro and in vivo [156]. It has been the subject of multiple clinical studies and trials, making it a unique therapeutic agent. Oral administration of Pycnogenol inhibited NFκB activity and reduced the secretion of MMP-9 in serum [157]. In a study involving 67 OA patients (34 control and 33 treatment group), the application of a Pycnogenol patch on the affected joint reduced the dependence on non-steroidal anti-inflammatory drugs and improved OA symptoms [158]. These findings were reaffirmed in another prospective, double-blind clinical trial in which 100 patients (50 in the control group and 50 in the treatment group) were randomly selected and given 150 mg Pycnogenol orally for three months. The study discovered that Pycnogenol improved and alleviated pain levels in the treatment group as determined by WOMAC scoring, and that the supplementation was well tolerated [159]. The active ingredients of Pycnogenol were found in the synovial fluid of OA patients administrated orally with pine bark extract suggesting that Pycnogenol active ingredient can reach the knee joint and show clinical efficacy [111]. In another randomized controlled trial, 33 OA patients (17 control and 16 treatment group) were given 100mg Pycnogenol orally, twice a day, for three weeks before knee arthroplasty surgery. Pycnogenol supplementation reduced the expression of cartilage ECM degrading proteases, MMP-3, and MMP-13 in the cartilage and ADAMTS5 levels in the serum [110]. These results were once again replicated in OA patients who experienced pain alleviation in a treadmill walk and improved joint function after 3-month treatment of Pycnogenol tablets. The same study reported a significant decrease in plasma free radical content, a key player in OA pathogenesis [160].

**4.16 Apigenin** was reported to inhibit the expression of MMP-1, -3, and -13 and ADAMTS-4 and ADAMTS5 in IL-1β stimulated rabbit chondrocytes [112]. Apigenin reduced cartilage degeneration and OA progression in a rat model of OA [112]. In another in vitro study using cartilage explants, Apigenin was reported to block IL-1β, TNF-α, and Oncostatin M-induced degradation of proteoglycan [161]. These studies show that plant polyphenols exert a chondroprotective effect in human cartilage explant, various animal models of OA, and randomized clinical trials.

**4.17 Icariin** is the major, effective natural compound isolated from Epimedium and has been shown in several studies to have chondroprotective activity [162]. Icariin suppressed NFκB signaling pathway in chondrocytes [114,163]. In a mouse ACLT model, intraarticular injection of Icariin suppressed Mmp-13 expression, increased Col2a1 expression, and reduced the severity of OA [113]. In another study, intraarticular administration of Icariin onto a poly(lactic-co-glycolic acid) (PLGA) scaffold suppressed the progression of OA in a rabbit model [115].

**4.18** The **terpene** class of natural compounds have been shown to influence cartilage metabolism. In IL-1β induced human chondrocytes, monoterpenes myrcene and limonene notedly reduced NFκB, JNK and p38 activation, decreased MMP-1 and MMP-13 expression, and inhibited nitric oxide production [116]. Myrcene exhibited the strongest effect on these mediators, also increasing expression of TIMP-1 and TIMP-3. Crocin, an active compound of the saffron spice, was seen to decrease gene expression of MMP-1, MMP-3, and MMP-13 via NFκB inhibition in ACLT-induced mouse model [117]. 

**4.19 Alkaloids** of note piperine which is a constituent of black pepper, was shown to significantly reduce gene expression of MMP-3, MMP-13, nitric oxide, and COX-2 in IL-1β induced human chondrocytes [70]. Although limited data are available on piperine, a clinical trial of 40 OA patients supplemented with an oral formulation of hyaluronic acid, chondroitin sulfate, keratin matrix, manganese and piperine saw significant reduction in pain and no side effects via WOMAC scoring [118].

**4.20** The **isoflavone** family has been studied extensively. Puerarin, found in the plant Pueraria lobata, was administered to ACLT-induced rats (at both 50 and 100 mg/kg daily dosages) and compared to a control group. The treatment group showed reduced protein expression of MMP-3, MMP-13, and ADAMTS5 while also targeting IL-1β, IL-6, and TNF-α and reversed type II collagen degradation [119]. Similar results of alleviating ECM degradation have been reproduced but hypothesized to occur through different mechanisms, such as inhibition of Nrf2/HO-1 and NFκB or upregulation of AMPK/PPAR-γ [164,165]. Genistein is another isoflavone which is derived from soybeans. In IL-1β induced human chondrocytes, Genistein reduced the protein expression of MMP-1, MMP-2, MMP-3, and MMP-13 as well as other catabolic factors such as nitric oxide, COX-2 through Nrf2 mediated inhibition of NFκB [166]. Another study found genistein treated MIA induced rats to present with significantly higher chondrocyte counts, but expression of catabolic proteins yielded mixed results [167].

## 5. Conclusions

In summation, this review discussed various plant derived natural compounds that prevent or reduce cartilage ECM catabolism/degradation in OA and promote ECM synthesis. Through the inhibition of the major inflammatory cytokine IL-1β, which is implicated in OA pathogenesis, the aforementioned compounds reduce the expression of downstream activators MMPs and ADAMTSs, while also increasing the expression of aggrecan and type II collagen and other key players. These results have been supported by in vitro and in vivo studies using chondrocyte culture, mouse, rat, rabbit, and guinea pig models and clinical trials (Table 2). Many of these bioactive compounds have a rich historical and cultural significance as therapeutic agents which predate modern clinical practice. Our review aims to underscore the utility of such compounds as potential therapeutic agents for OA. A thorough understanding of the biological process and molecular regulation of MMP-13 and ADAMTS5, combined with the current stages of development of various natural compounds, may lead to the identification of future therapeutic strategies. Based on the current studies on several of the natural bioactive compounds and their bioavailability in the synovial joints and their anti-inflammatory, antioxidant, and chondroprotective activity and favorable safety profile, these compounds have the potential to be added as supplements for OA patients. Future directions may include the study of the clinical efficacy of the reported natural compounds, with an emphasis on synergism with other compounds, the optimal route of administration, and adverse effects.

## Figures and Tables

**Figure 1 life-13-00102-f001:**
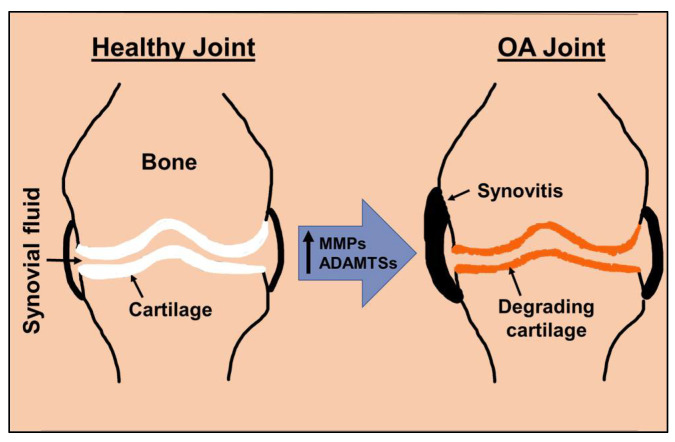
Schematic representation of normal and OA joints depicting cartilage matrix degradation.

**Figure 2 life-13-00102-f002:**
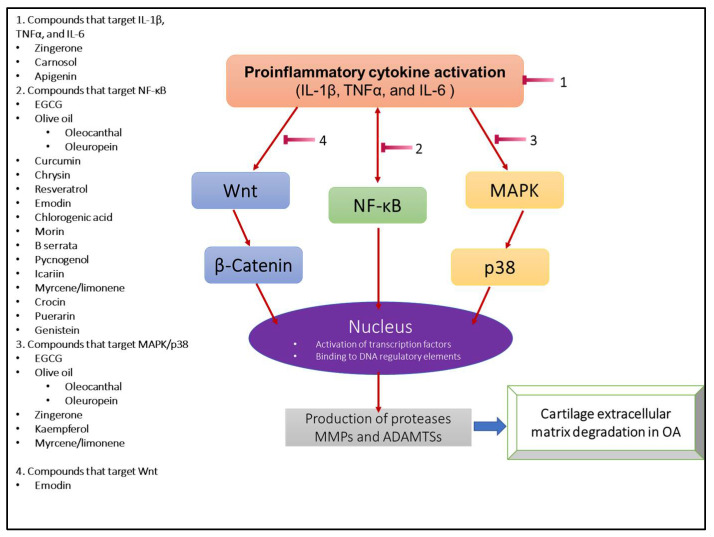
Schematic representation of the pathological signaling pathways in OA triggering the expression of matrix-degrading proteases. The list of the natural compounds on the left have been shown to have the potential to be developed as therapeutics for the treatment of OA.

**Table 1 life-13-00102-t001:** List of bioactive natural compounds, the model system of study and their targets.

Compound	Model	Target/Action	Reference
Butein	Human chondrocyte	Suppression of MMP-3, MMP-9, MMP-13	[64,65]
EGCG	AGEs stimulated human chondrocyte, MIA rat, DMM mouse, guinea pig	Suppression of MMP-1, MMP-13, and ADAMTS5	[71,72,73,74,75]
Olive oil			
Hydroxytyrosol	Surgically induced rabbit and mouse	Suppression of MMP-13 and protection of aggrecan via antioxidant activity	[76]
Oleocanthal	LPS-induced human chondrocyte	Significant suppression in MMP-13, and ADAMTS5 via MAPK/ NFκB inhibition	[77]
Oleuropein	Human chondrocyte, guinea pig	Suppression of MMP-1, MMP-13, ADAMTS5, and increased deposition of Col2A1/proteoglycan via MAPK/NFκB inhibition	[78,79,80]
Curcumin	Human cartilage explant, rat primary chondrocyte, zymosan mouse, DMM mouse, equine cartilage explant	Suppression of MMP-3, MMP-8, MMP-13, ADAMTS5 via NFκB inhibition and increased expression of type II collagen and CITED2.	[81,82,83,84,85,86]
Honey			
Chrysin	Human chondrocyte	Suppression of MMP-1, MMP-3, MMP-13, ADAMTS5 via NFκB inhibition and reduced HMGB-1 activity	[87]
Fisetin	Human chondrocyte, rat	Decreased MMP-3, MMP-13, ADAMTS5 expression	[88,89]
Resveratrol	Human chondrocyte, ACLT rabbit, DMM mouse, porcine cartilage explant	Suppression of MMP-13 via JNK/ERK-AP-1 inhibition, suppression of MMP-1, MMP-3, MMP-13, ADAMTS4, ADAMTS5 via NFκB inhibition, increased expression of type II collagen and aggrecan via AMPK/mTOR signaling, activation of SIRT1 and inhibition of HIF-2α	[90,91,92,93]
Zingerone	Human cartilage explant	Suppression of MMP-13 via p38/JNK-MAPK pathway	[94]
Kaempferol	Human chondrocyte, rat chondrocyte	Suppression of MMP-1, MMP-3, MMP-13, ADAMTS4, ADAMTS5 via p38/ERK-MAPK inhibition, suppression of STAT3, inhibition of type II collagen degradation	[95,96]
Emodin	Human chondrocyte, rat chondrocyte, ACLT rat	Suppression of MMP-3, MMP-13, ADAMTS4, ADAMTS5 via NFκB and Wnt/B-catenin inhibition, preservation of aggrecan and type II collagen	[97,98]
Carnosol	Human chondrocyte	Suppression of MMP-3, ADAMTS4, ADAMTS5, increased expression of type II collagen and aggrecan	[99]
Ferulic acid	Human chondrocyte, papain rat	Suppression of MMP-1, MMP-3, MMP-13 via SIRT1/AMPK/PGC-1α inhibition, restoration of SOX9, upregulation of TIMP-1	[100,101,102]
Chlorogenic acid	Human chondrocyte, human cartilage explant, rabbit chondrocyte, ACLT rabbit, rat chondrocyte	Suppression of MMP-1, MMP-3, MMP-13, ADAMTS4, ADAMTS5 via NFκB inhibition, increased expression of type II collagen and aggrecan	[103,104]
Quercetin	Rat chondrocyte	Suppression of MMP-1, MMP-3, MMP-9	[105,106,107]
Morin	Human chondrocyte, ACLT rat	Suppression of MMP-3, MMP-13 and upregulation of TIMP-1	[108]
B serrata	Human chondrocyte, human cartilage explant	Suppression of MMP-9, MMP-13	[109]
Pycnogenol	Human chondrocyte	Suppression of MMP-3, MMP-9, MMP-13, ADAMTS5 via NFκB inhibition	[110,111]
Apigenin	Human cartilage explant, rabbit chondrocyte, rat chondrocyte	Blocking IL-1β, TNF-α, and suppression of MMP-1, MMP-3, MMP-13, ADAMTS4, ADAMTS5	[112]
Icariin	Rabbit chondrocytes and OA model, mouse OA model	Suppression of Mmp-13 and increased Col2a1 levels by targeting Indian Hedgehog and NFκB pathway	[113,114,115]
Terpenes			
Myrcene/ Limonene	Human chondrocyte	Reduced nitric oxide production, induced TIMP-1 and TIMP-3 expression, suppressed MMP-1 and MMP-13 expression via NFκB, JNK, and p38 inhibition	[116]
Crocin	ACLT mouse	Suppression of MMP-1, MMP-3, MMP-13 gene expression via NFκB inhibition	[117]
Alkaloids			
Piperine	Human chondrocyte	Suppression of gene expression of MMP-3, MMP-13, nitric oxide, COX-2	[70,118]
Isoflavones			
Puerarin	ACLT rat, MIA rat, human chondrocyte,	Suppression of MMP-3, MMP-13, ADAMTS5 protein expression, reducing levels of IL-1β, IL-6, TNF-α, and reversing type II collagen degradation via Nrf2/HO-1 and NFκB inhibition	[119]
Genistein	Human chondrocyte, MIA rat	Suppression of MMP-1, MMP-2, MMP-3, MMP-13 protein expression via Nrf-2 mediated NFκB inhibition	

**Table 2 life-13-00102-t002:** Summary of clinical trials.

Reference	Sample Size	Supplementation	Results
[124]	n = 20	8 caps, 400 mg each, of grapeseed and olive extract (hydroxytyrosol and procyanidins content)	Venous blood collected post-ingestion showed peak metabolic concentration at 100 min with reduction in IL-1β and inflammatory cytokines
[79]	n = 124 (62 control, 62 treatment)	One capsule of 50 mg oleuropein twice a day	Knee injury and Osteoarthritis Outcome Score (KOOS) determined significantly reduced walking pain in subjects
[131]	n = 22	6 caps, 42 mg each, of bio-optimized curcumin per day	Significant reduction in Coll2-1 and insignificant pain alleviation
[140]	n = 110 (55 control, 55 treatment)	15 mg meloxicam + 500 mg resveratrol once daily	Western Ontario and McMaster Universities Osteoarthritis Index (WOMAC) scores indicated significant improvement in pain, stiffness, and physical function
[141]	n = 82 (32 control, 50 treatment)	15 mg meloxicam + 500 mg resveratrol once daily	WOMAC scoring indicated no significant clinical relief with minimal difference in IL-1β, TNF-α, and IL-6 serum level
[145]	n = 120 (60 control, 60 treatment)	One capsule of 500 mg powdered ginger twice daily	At 3 months, there was a significant reduction in IL-1β and TNF-α concentrations in the treatment group
[146]	n = 120 (60 control, 60 treatment)	One capsule of 500 mg powdered ginger twice daily	At 3 months, there was a significant reduction in serum nitric oxide and hs-C reactive protein levels in the treatment group
[147]	n = 68 (34 control, 34 treatment)	Knee massage with ginger oil twice a week	WOMAC scoring and visual analog scale (VAS) determined significant clinical relief in pain, stiffness, and function in treatment group
[96]	n = 99 (33 control, 33 low-dose, 33 high-dose)	Low-dose: 300 mg of *Elaeagnus angustifolia* extract with kaempferol administered as syrup in two doses per day High-dose: 600 mg of *Elaeagnus Angustifolia* extract with kaempferol administered as syrup in two doses per day	WOMAC, VAS, and Leguesne’s Pain-Function Index (LPFI), and Patient’s Global Assessment (PGA) all indicated improvement for both dosages after 7 weeks. Low and high dosages exhibited significant reduction in pain and stiffness while only high dose exhibited improvement in physical function
[106]	n = 40 (20 control, 20 treatment)	6 tablets of 1200 mg glucosamine hydrochloride, 60 mg chondroitin sulfate and 45 mg quercetin glycosides per day	After 16 weeks, treatment group experienced pain alleviation with walking and ascending/descending the stairs, per Japan Orthopaedic Association (JOA) criteria. Type II collagen levels were preserved, although not significant
[158]	n = 67 (34 control, 33 treatment)	Pycnogenol patch was applied to affected joint	Treatment group experienced reduced dependence on non-steroidal anti-inflammatory drugs, improved OA symptoms, and significant reduction in C-reactive protein (CRP) and erythrocyte sedimentation rate (ESR)
[159]	n = 100 (50 control, 50 treatment)	150 mg Pycnogenol per day with meals	WOMAC and VAS criteria determined the treatment group experienced significant reduction in pain by the first month, while maximum effect was seen by the second month
[111]	n = 33 (17 control, 16 treatment)	2 capsules of Pycnogenol, 50 mg each, twice daily	Supplementation was well tolerated and distributed into the synovial fluid of OA patients
[110]	N = 33 (17 control, 16 treatment)	100 mg Pycnogenol twice a day	Treatment group saw reduced expression of IL-1β, MMP-3, MMP-13, and ADAMTS5 levels in the serum after 3 weeks
[160]	n = 55 (26 control, 29 treatment)	2 tablets, 50 mg Pycnogenol each, per day	After 3 weeks, the treatment group experienced significant reduction in CRP levels and plasma free radicals
[118]	n = 40	Oral formulation including piperine given over one month span	After 2 months, participants experienced significant reduction in pain via WOMAC scoring and no side effects/good tolerability

## Data Availability

Not applicable.

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
