# Peer review of "Natural Compounds: Potential Therapeutics for the Inhibition of Cartilage Matrix Degradation in Osteoarthritis"

_life, 2022, doi:10.3390/life13010102_

Round 1

Reviewer 1 Report

This is review describes the possible role of 'natural' compounds for the inhibition of matrix breakdown as suggested to occur in OA. 

However it is really important that the authors acknowledge that the majority of studies have been preformed on animal models and in vitro experiments so there should be a strong cautionary note about the extension of these studies to be useful therapeutic strategies for human idiopathic (i.e. Primary) OA. 

Specific points:

Line 43 – not accurate to state that chondrocytes are the only cell type.

Williams, R., et al  (2010). Identification and clonal characterisation of a progenitor cell subpopulation in normal human articular cartilage. PLoS One, 5(10), e13246. https://doi.org/10.1371/journal.pone.0013246

Line 43 – stating ‘water’ over-simplifies to constituents of the interstitial fluid - please also describe constituents accurately.

MMPs & ADAMTs – involved in matrix turnover. Note that for this treatment approach to be effective, maintaining or even increasing levels of matrix synthesis would be required.

Line 66 -67. Too strong as link to preventing human OA progression not proven.

Para starting line 68 – comment on ADAMTs-4 should be included.

Figure 1 legend –‘..have a protective effect on OA….’ This is not accurate and potentially misleading.

Line 289 – ADAMTs-4

Lines 358 - what evidence was presented that these 'natural componds' 'promoted' ECM synthesis?

Line 373 Unwise and inaccurate to state that these'...compounds can be added as an add-on supplement for OA patients'. No clinical evidence in support of this statement.

Are there any authoritative clinical studies that support a possible role for these products in preventing/retarding OA progression?

Author Response

Reviewer 1.

However it is really important that the authors acknowledge that the majority of studies have been preformed on animal models and in vitro experiments so there should be a strong cautionary note about the extension of these studies to be useful therapeutic strategies for human idiopathic (i.e. Primary) OA. 

Response. We have added more clinical studies in the revised version of the manuscript. We have added a sperate table for clinical studies in the revised version of the manuscript.

Specific points:

Line 43 – not accurate to state that chondrocytes are the only cell type.

Williams, R., et al  (2010). Identification and clonal characterisation of a progenitor cell sub‐population in normal human articular cartilage. PLoS One, 5(10), e13246. https://doi.org/10.1371/journal.pone.0013246

Response. Thank you to the reviewer for his comment. We have changed from “only cell type” to “primary cell type” in the revised manuscript.

Line 43 – stating ‘water’ over-simplifies to constituents of the interstitial fluid - please also describe constituents accurately.

Response.  Thank you for pointing it out. We have modified the sentence as “The cartilage ECM is composed of a complex network of biological macromolecules that exist in an aqueous environment that allows nutrients, growth factors, and cytokines to move freely between chondrocytes and synovial fluid” in the revised manuscript and have added additional references.

MMPs & ADAMTs – involved in matrix turnover. Note that for this treatment approach to be effective, maintaining or even increasing levels of matrix synthesis would be required.

Response. This is a very good point raised by the reviewer and we agree with his view. In addition to stop the matrix degradation, an ideal drug/compound is expected to increase the matrix synthesis to make up the loss caused by degradation. We acknowledge this in our manuscript, line 12-13 and 38-40.

Line 66 -67. Too strong as link to preventing human OA progression not proven.

Response. We have revised the manuscript and changed the wordings in the paper as “The results from these animal studies demonstrate that MMP-13 is a critical player in OA pathogenesis suggesting that MMP-13 inhibition may reduce disease severity. ”

Para starting line 68 – comment on ADAMTs-4 should be included.

Response. We have added ADAMTS4 in the revised manuscript, however, since it has been shown that ADAMTS5 plays a major role in OA pathogenesis (PMID: 17968948, 15334469, 15800624), when compared to ADAMTS4, we have focused mainly on ADAMTS5. The above references have been added in the revised manuscript to justify the focus on ADAMTS5.

Figure 1 legend –‘..have a protective effect on OA….’ This is not accurate and potentially misleading.

Response. We have modified the legend of the figure in the revised manuscript. The new legend is “Schematic representation of the pathological signaling pathways in OA triggering the expression of matrix-degrading proteases. The list of the natural compounds on the right have been shown to have the potential to be developed as therapeutics for the treatment of OA.”

Line 289 – ADAMTs-4

Response. We have added ADAMTS4 in the revised manuscript.

Lines 358 - what evidence was presented that these 'natural componds' 'promoted' ECM synthesis?

Response. Studies with some of the natural compounds were shown to increase the synthesis of the ECM components. Oleuropein improved collagen and aggrecan synthesis; line 220-222. In a clinical study, Quercetin improved aggrecan synthesis; line 329-331.

Line 373 Unwise and inaccurate to state that these'...compounds can be added as an add-on supplement for OA patients'. No clinical evidence in support of this statement.

Are there any authoritative clinical studies that support a possible role for these products in preventing/retarding OA progression?

Response. This is a very good point and an important concern. The in vitro or animal studies with natural compounds show a promising result. In clinical trials the results are supportive or inconclusive but at least there are no side effects reported. Most of the studies reported short term beneficial effects but no long-term benefit compared to placebo (reviewed by Xiaoqian et al in Br J Sports Med, 2018 Feb;52(3):167-175; PMID:29018060).  In one of the metanalysis article on curcumin by Hsiao et al (Complement Ther Med. 2021 Dec;63:102775. doi: 10.1016/j.ctim.2021.102775) that included 11 studies with 1258 participants concluded that curcumin can be recommended as adjunctive. We have included above reference in our review. Based on the above references, and suggestion from the reviewer, we have modified the sentence as “Based on the current studies on several of the natural bioactive compounds and their bio-availability in the synovial joints and their anti-inflammatory, antioxidant, and chondro-protective activity and favorable safety profile, these compounds have the potential to be added as supplements for OA patients”

Reviewer 2 Report

In general, the topic is interesting but there are a lot of review published. Thus, the review should be improved to add something new and to distinguish it from other papers already published.

I have several comments for the authors.

The introduction on OA should be improved. OA is a whole joint disease involving all joint tissues including meniscus and infrapatellar fat pad.

Lines 41-48: references should be added.

In the introduction the authors focused on cartilage, chondrocytes and cartilage ECM. However, it was not mentioned the pericellular matrix and the chondrons.

Lines 57-58: MMPs are produced also by synovial membrane and meniscus. It should be added with relative references.

At the end of the introduction, the aim of the review should be better explained. Indeed, the authors focused also on “Matrix degradation is central to osteoarthritis” and “Signaling pathways regulating matrix degradation in osteoarthritis”.

Figures could be added for section 2 and 3.

Lines 190-112: also in synovial membrane.

Lines 112-115, lines 287-288: references should be added.

Line 126: here the authors reported “a focus of this review is NFκB signaling”. Again, the aim/focus of the review should be clarified.

Line 161 and figure 1: the authors focused only on flavonoids but the review and the title are related to natural compounds and not flavonoids.

Icariin, a type of flavonoid, was not mentioned.

There are other relevant natural compounds that were not discussed at all by the authors. Terpenes (i.e. β-caryophyllene, crocin etc), alcaloids (i.e piperine), isoflavones ( such as puerarin, genistein) were studied in the context of OA and cartilage/chodnrocytes.   

Line 165: Is there a reason why butein-rich is bolded?

Table 1: layout should be improved. The content should be improved. For example, the model used and the targets should be better explained. The authors could consider to report one table for studies performed in vitro and one for studies in vivo.

A table reporting the results of clinical trials would be useful.

Abbreviations should be defined at first mention.

Author Response

Reviewer 2.

In general, the topic is interesting but there are a lot of review published. Thus, the review should be improved to add something new and to distinguish it from other papers already published.

I have several comments for the authors.

The introduction on OA should be improved. OA is a whole joint disease involving all joint tissues including meniscus and infrapatellar fat pad.

Response. We have included these points in the revised manuscript.

Lines 41-48: references should be added.

Response. Additional references have been added in the revised manuscript as suggested by the reviewer.

In the introduction the authors focused on cartilage, chondrocytes and cartilage ECM. However, it was not mentioned the pericellular matrix and the chondrons.

Response.

Lines 57-58: MMPs are produced also by synovial membrane and meniscus. It should be added with relative references.

Response. We have revised this part as suggested by the reviewer and have added additional reference

At the end of the introduction, the aim of the review should be better explained. Indeed, the authors focused also on “Matrix degradation is central to osteoarthritis” and “Signaling pathways regulating matrix degradation in osteoarthritis”.

Response. We are thankful to the reviewer for this comment. This will improve the quality of the manuscript and give the reader a better understanding of the topic discussed in this article. We have revised the last paragraph of the introduction to better explain the focus of this review article.

Figures could be added for section 2 and 3.

Response. The figure 2 in the revised manuscript represents both section 3 and section 4. We have added an additional figure (Figure 1) in the revised manuscript for section 2.

Lines 190-112: also in synovial membrane.

Response. We have revised the manuscript as suggested by the reviewer.

Lines 112-115, lines 287-288: references should be added.

Response. We have added references at these places.

Line 126: here the authors reported “a focus of this review is NFκB signaling”. Again, the aim/focus of the review should be clarified.

Response. We have modified the sentence in the revised manuscript to stay focused on the topic. The modified sentence is “One of the most important pathways activated in OA and discussed in this review is NFκB signaling”

Line 161 and figure 1: the authors focused only on flavonoids but the review and the title are related to natural compounds and not flavonoids.

Response. Thank you for pointing it out. We have revised the manuscript. In the last paragraph of the introduction, we clarify that the focus of this manuscript is the role of plant derived natural compounds in OA.

Icariin, a type of flavonoid, was not mentioned.

Response. We have added Icariin in the revised manuscript.

There are other relevant natural compounds that were not discussed at all by the authors. Terpenes (i.e. β-caryophyllene, crocin etc), alcaloids (i.e piperine), isoflavones ( such as puerarin, genistein) were studied in the context of OA and cartilage/chodnrocytes.  

Response. We have mentioned these compounds in the revised manuscript.

Line 165: Is there a reason why butein-rich is bolded?

Response. We have corrected it in the revised manuscript. It is not bold.

Table 1: layout should be improved. The content should be improved. For example, the model used and the targets should be better explained. The authors could consider to report one table for studies performed in vitro and one for studies in vivo.

A table reporting the results of clinical trials would be useful.

Response. We have revised the table in the resubmitted manuscript. As suggested by the reviewer, we have added a separate table of clinical trial studies in the revised manuscript. Thank you for this suggestion.

Abbreviations should be defined at first mention.

Response. Thank you for pointing it out. We have checked the manuscript for any abbreviations and made sure that it is defined at first use.

Round 2

Reviewer 1 Report

The authors have taken careful consideration of this referee's comments and the manuscript has been improved. The authors should note the continuing debate about the relative importance of ADAMTs-4 and ADAMTs-5. The authors should note that the referencs used (PMID: 17968948, 15334469, 15800624) to support a primary role for ADAMTs-5 are from animal and in vitro studies (lines 78-81). The authors should be very cautious about extending animal studies of spontaneous/induce OA to those for human clinical idiopathic OA - a more critical assessment would be appropriate. Note also that the study of Bau et al 2002 measured expression (of ADAMTs-4 and -5) and not protein levels. Small changes (e.g. in ADAMTs-4) might be considered insignificant but not when they affect critical cell populations in cartilage - e.g. those in the superficial zone.

Note the correction required on line 396 - ADAMTs

Author Response

Reviewer 1

The authors have taken careful consideration of this referee's comments and the manuscript has been improved. The authors should note the continuing debate about the relative importance of ADAMTs-4 and ADAMTs-5. The authors should note that the referencs used (PMID: 17968948, 15334469, 15800624) to support a primary role for ADAMTs-5 are from animal and in vitro studies (lines 78-81). The authors should be very cautious about extending animal studies of spontaneous/induce OA to those for human clinical idiopathic OA - a more critical assessment would be appropriate. Note also that the study of Bau et al 2002 measured expression (of ADAMTs-4 and -5) and not protein levels. Small changes (e.g. in ADAMTs-4) might be considered insignificant but not when they affect critical cell populations in cartilage - e.g. those in the superficial zone.

Response. This is a very good point. We have modified the manuscript in the abstract and introduction section to talk about both Adamts4 and Adamts5.

Note the correction required on line 396 – ADAMTs

Response. Thank you for pointing out the typo. We have corrected it in the revised manuscript.

Reviewer 2 Report

In my previous report, I asked to the authors: “In the introduction the authors focused on cartilage, chondrocytes and cartilage ECM. However, it was not mentioned the pericellular matrix and the chondrons.” However, I was not able to find any reply by the authors to this comment.

The authors decided to maintain the focus the review on natural compounds but the review is still unbalanced towards to flavonoids. The authors should add and discuss the main terpenoids and alkaloids at least or the aim/title of the review must be modified. Reporting only a sentence is not enough (lines 181-183).  Figure 2 and table 1 are still focused on flavonoids.

Author Response

Reviewer 2

In my previous report, I asked to the authors: “In the introduction the authors focused on cartilage, chondrocytes and cartilage ECM. However, it was not mentioned the pericellular matrix and the chondrons.” However, I was not able to find any reply by the authors to this comment.

Response. it was an inadvertent mistake. We have added chondrons in the introduction section, line 47-48.

The authors decided to maintain the focus the review on natural compounds but the review is still unbalanced towards to flavonoids. The authors should add and discuss the main terpenoids and alkaloids at least or the aim/title of the review must be modified. Reporting only a sentence is not enough (lines 181-183).  Figure 2 and table 1 are still focused on flavonoids.

Response. We have added terpenes, alkaloids and isoflavones and have updated the figure 2, table 2 in the revised manuscript. Section 4.18, 4.19 and 4.20.